

# Enhanced mutation strategy based differential evolution for global optimization problems

Pawan Mishra[1], Musrrat Ali[2], Pooja[1] and Safiqul Islam[2]

[1] Department of Electronics and Communication, University of Allahabad, Prayagraj, India
[2] Department of Basic Sciences, King Faisal University, Al Ahsa, Saudi Arabia

## ABSTRACT

Differential evolution (DE) stands out as a prominent algorithm for addressing global optimization challenges. The efficacy of DE hinges crucially upon its mutation operation, which serves as a pivotal mechanism in generating diverse and high-quality solutions. This article explores various mutation operations aimed at augmenting the performance of DE in global optimization tasks. A distinct mutation strategy is introduced, with the primary objective of achieving a harmonious equilibrium between exploration and exploitation to enhance both convergence speed and solution quality. The proposed DE centres on a novel mutation-based strategy, introducing a new coefficient factor ("σ") in conjunction with the base vector of the basic mutation strategy ("DE/rand/1"). This innovation aims to fortify the convergence of local variables during exploitation, thereby improving both the convergence rate and quality. The effectiveness of the proposed mutation operations is evaluated across a set of 27 benchmark functions commonly employed in global optimization. Experimental results conclusively demonstrate that these enhanced mutation strategies significantly outperform state-of-the-art algorithms in terms of solution accuracy and convergence speed. This study underscores the critical role of mutation operations in DE and provides valuable insights for designing more potent mutation strategies to tackle complex global optimization problems.

## INTRODUCTION

The present study shows that differential evolution (DE) is continuously becoming the most dominant optimization algorithm to efficiently handle the complex mathematical problems. With such qualities, it is frequently used in real-world problems like share-market prediction analysis, making efficient production processes and providing quality products, finding the optimum performance of software applications with the utilization of minimum resources and less time, finding out the equilibrium state of chemical reactions, and many more. The DE, which comes under evolutionary algorithms, was first introduced by an American and a German researcher, namely Rainer Stron and Kanneth Price, in 1995 (*Price, 1996*). Essentially, it addressed the Chebyshev mathematical polynomial problems. Later, the new variant of DE was proposed to efficiently handle the numerical optimization problem over continuous search space optimization in 1997

Corresponding authors
Musrrat Ali, mkasim@kfu.edu.sa
Pooja, cs.pooja@allduniv.ac.in

[1] Portions of this text were previously published as part of a pre-print (*Mishra et al., 2024*)

(*Storn & Price, 1997*). Consequently, DE came into focus and became the most curious and dominant area for researchers and scientists. DE follows the heuristic method to achieve its goal (*Price, 2013*)[1].

The basic conceptual phenomenon behind DE is the same as in the other evolutionary algorithms (*Back, 1996*), including Darwin's biological evolution theory, "survival of the fittest." It means that the candidate solution with more potential will go through the evolutionary process of the next generation, and finally, the most suitable candidate will survive at the end. DE involves some biologically inspired operators like mutation, crossover, and selection, each of which has its own significance. There are two control parameters (mutant scaling *F* and crossover probability factor *Cr*), that are involved in mutation and crossover processes, respectively. Sometimes, the minor tuning of these control factors might produce excellent results. Therefore, it has a direct impact on the performance of DE. Many researchers are still working on this aspect (adjusting the control parameter) (*Eiben, Hinterding & Michalewicz, 1999*; *Karafotias, Hoogendoorn & Eiben, 2015*; *Liu et al., 2017*), and continuously producing good results. Overall, parameter setting plays a crucial role in problem convergence, which is evident in the diversity of the population.

Along with the control parameter, the performance of DE also depends on the size and structure of the population (*Lynn, Ali & Suganthan, 2018*; *Yang et al., 2015*). In a larger population (*Zhu, Tang & Zhang, 2013*), it might be hard to find the best solution and there is a smaller chance of finding the right search direction as the algorithm is improved over many generations. In the literature, many researchers have shown that these are the crucial factors that have enough potential to have a valuable impact on the performance of the DE algorithm.

Therefore, researchers might be more concerned about the selection of these prime factors, and they should always ensure a good balance between them. If this section is not perfect enough, it might affect the exploitation-exploration mechanism and may cause the premature convergence problem as well. To overcome such challenges, DE evolves the population over several generations in order to achieve the desired outcome (*Mallipeddi et al., 2011*).

The rest of the article has been organized as follows:

"Literature Review" provides the detailed literature review with the latest research articles; "Differential Evolution Algorithm" explores the Differential Evolution algorithm; the "Proposed Differential Evolution Algorithm (EMDE)" describes the proposed approach used in this article; "Standard Benchmark/Fitness Function" gives information about the standard benchmark/fitness function; "Comparative Study and Discussion" provides experimental results and discussion, and finally section seventh gives the concluding remarks of proposed.

## LITERATURE REVIEW

For many decades, although there are many algorithms already developed, the premature convergence of populations is still a big challenge for researchers. *Brest et al. (2006)* presented the jade Differential Evolution Algorithm (jDE) algorithm with a parameter-

adaptive strategy for constraint optimization problems (*Brest et al., 2006*). The "$current - P_{best}/1$" strategy and an adaptive parameter were used in the JADE algorithm for adaptive differential evolution with optional external archive (*Zhang & Sanderson, 2009*). Self-Adaptive Differential Evolution (SaDE) has been proposed with an adaptive difference approach (*Qin, Huang & Suganthan, 2009*). Cooperative Differential Evolution (CoDE) used the trail vector strategy along with the setting of control parameters and randomly combined them with each of three different variants, respectively, and finally generated a new trail vector (*Wang, Cai & Zhang, 2011*). Enhanced Population-based Self-adaptive Differential Evolution (EPSDE) was proposed with a new mutation strategy along with a control parameter approach (*Mallipeddi et al., 2011*). Controlled Population Differential Evolution (CPDE) was proposed with a self-adaptive nature of control parameter, and Controlled Population Multi-objective Differential Evolution (CPMDE) is an enhanced version of Controlled Population Differential Evolution (CPDE), where a modified mutation strategy has been embedded with a self-tuned control parameter approach (*Pooja, Chaturvedi & Kumar, 2015*).

There are several variants proposed based on the crossover operator. Orthogonal Differential Evolution (ODE) was proposed with the Orthogonal Crossover (OX) based on the orthogonal concept to widely explore the search space (*Gong, Cai & Ling, 2006*). A new method called "CoBiDE," which is based on covariance-based matrix learning, has been suggested for a crossover operator ($Cr$) that aims to explore and use the search space (*Wang et al., 2014*). Success history-based control parameter adaptation strategy has been used for further generation (*Tanabe & Fukunaga, 2013*). A novel differential evolution (NDE) algorithm was proposed with a dual strategy (a self-adaptive approach to control parameters and a single population structure) (*Pooja et al., 2018*).

Moreover, there are many variants of DE proposed on the basis of population structure. Multi-Parent Differential Evolution (MPEDE) was proposed with a multiple population structure (derived from a single base population) along with a selected approach as a strategy set (*Wu et al., 2016*). The innovative hybridisation of the whale optimisation method and the differential evolution algorithm, termed the novel hybridised whale-differential evolution optimisation algorithm, is designed for engineering design problems (*Mishra, Pooja & Tripathi, 2024*). Differential evolution with population segment tuning, (DEPT), is an approach of DE having population segmentation (*Pooja, Chaturvedi & Kumar, 2016*). Muti-Cooperative Differential Evolution (MCDE) was proposed with a multi-population mechanism for three different mutation strategies along with a co-variance approach for crossover operation (*Du et al., 2019*), while multi-population differential evolution with a balanced ensemble of mutation strategies was delivered by *Ali, Awad & Suganthan (2015)*. An improved differential evolution algorithm based on the dual-strategy-based differential evolution (DSIDE) algorithm has been introduced with a dual-strategy-based approach in 2022, where an enhanced mutation strategy along with a scaling-based self-control parameter has been applied (*Zhong & Cheng, 2020*).

Although there are many variants of DE available, the population stagnation and quality convergence still remain as challenges. To improve the quality of convergence, researchers have proposed and validated a new mutation strategy across numerous standard fitness

functions. Finally, this novel approach outperforms other DE variants in many aspects. The proposed method can be applied to a variety of real-life problems (*Rosas-Caro et al., 2021*; *Boursianis et al., 2021*; *Lotfi, Mardani & Weber, 2021*; *Pooja, 2023*; *Yu, Wu & Luo, 2022*).

The differential evolution is not only limited to optimized the mathematical objective and common real-life problems, but it is also dealing with some advanced technologies like neural networks, quantum computing, *etc.* The SaDENAS, a self-adaptive differential evolution algorithm designed to optimize neural architecture search, enhancing model performance through efficient search strategies in evolving neural network structures (*Han et al., 2024*). An enhanced adaptive differential evolution algorithm has been proposed that uses dual performance evaluation metrics to improve the effectiveness of numerical optimization by balancing exploration and exploitation strategies in search processes (*Tian, Yan & Gao, 2024*). An improved differential evolution algorithm utilizing cooperative multipopulation strategies to improve optimization performance by fostering collaboration among diverse populations to explore solutions more effectively (*Shen et al., 2024*).

The authors aim to improve financial decision-making by effectively balancing risk and return, leveraging advanced optimization techniques for better performance in complex investment scenarios. The distributed nature of the algorithm improves its scalability and efficiency in solving the real world portfolio challenge (*Song et al., 2023*).

In 2023, researchers proposed a differential evolution algorithm that incorporates leader-adjoint populations, thereby enhancing the exploration and exploitation of the search space. The proposed approach improves optimization efficiency and solution quality across various problems, demonstrating significant performance gains in computational experiments compared to traditional differential evolution methods (*Li et al., 2023b*).

A differential clustering evolution technique using a neighbourhood-based dual mutation operator targeted for multimodal multi-objective optimization. The approach enhances solution diversity and convergence by effectively exploring multiple optimal solutions, demonstrating improved performance on complex optimization tasks compared to traditional methods through extensive experimental validation (*Zhou et al., 2023*). A novel evolving operator selector for the differential evolution algorithm that leverages fitness landscape information, by adapting the selection of mutation operators based on landscape characteristics, the approach enhances optimization performance and adaptability, leading to more effective exploration and exploitation of solution spaces in various problem domains (*Li et al., 2023a*).

The comprehensive review explores the applications of differential evolution in image processing problems. It examines various methodologies, strengths, and challenges of using this optimization technique in tasks such as image segmentation, enhancement, and feature extraction. The authors also highlight future research directions and potential improvements in the field (*Chakraborty et al., 2023*).

The comparative analysis of particle swarm optimization and differential evolution, evaluating their performance across various optimization tasks. It highlights strengths and

weaknesses, providing insights into the suitability of each algorithm for different applications (*Piotrowski, Napiorkowski & Piotrowska, 2023*). A joint differential evolution algorithm specifically designed for a reconfigurable intelligent surface (RIS)-assisted multi-UAV Internet of Things (IoT) data collection system has been proposed in 2024. To optimizing resource allocation and communication strategies, the algorithm improves data transmission efficiency and network performance, addressing challenges in complex aerial network environments and enhancing overall system functionality (*Li et al., 2024*). The authors introduce a dynamic dual-population differential evolution algorithm designed to tackle numerical and engineering optimization challenges. By utilizing two adaptive populations, leading to improved solution quality and faster convergence rates. Extensive experiments demonstrate its effectiveness across various optimization scenarios (*Zuo & Gao, 2024*).

## Contributions

The valuable contributions of this research article are mentioned below:

1. A novel edge strategy has been proposed to optimize the complex mathematical problems. It utilizes the mutation strategy with respect to mathematical formulation change for getting the best outcome over the targeted problem.
2. The proposed method is extensively evaluated on various types of benchmark functions and compared against many state-of-the-art algorithms. The experimental evaluation demonstrates the effectiveness of the proposed method on most of the benchmark functions. The proposed method outperforms the other approaches and is more accurate in getting the possible converged results.

## DIFFERENTIAL EVOLUTION ALGORITHM

DE is a most emerging algorithm, which works on real value parameter optimization. DE follows a greedy approach to reaching its goal.

DE could be applied over several types of function optimization:

Let $f(x) \in R$; therefore, $f^*(x) \in R$

where $f(x)$ represents the random real value of the population set, and $f^*(x)$ represents the optimized candidate solution, which must belong to real value.

DE consists of four main phases in the algorithm: population initialisation, mutation, crossover, and selection.

### Population initiation

This is the initial stage to establish the population within a continuous search space of defined dimensionality. DE produces a randomly dispersed population in the format of actual values. The proposed solution is presented below:

$$X_i^G, (i = 1, 2, \ldots, N_\rho), \tag{1}$$

where the population comprises $N\rho$ candidate solutions throughout a generation 'G'. It consistently produces a novel base population $X_i$ for analysis across multiple generations

$G_1, G_2, \ldots, G_{max}$. $G_{max}$ denotes the maximum generation engaged in the overall algorithmic process.

## Mutation

The mutation strategy is the crucial part of the overall DE algorithm, as it is responsible for converging the population structure. Each execution of the mutation process produces a unique and distinct population. The mathematical expression for mutation is formulated to generate a mutant population that retains the same structure as the beginning population. Three separate random indices $(\alpha_1, \alpha_2, \alpha_3)$ are selected from the initial population, and these random candidate solutions are utilised to construct a mutant.

The mathematical representations of various mutation techniques are presented here.

$$\text{"}DE/rand/1\text{"} : Y_i^G = X_{\alpha_1}^G + F * (X_{\alpha_2}^G - X_{\alpha_3}^G), \tag{2}$$

where, $rand/1$ indicate the mutation strategy. It is first ever proposed and most frequently used mutation strategy. Here, the base vector $(X_{\alpha_1}^G)$ added with the difference vector $(X_{\alpha_2}^G - X_{\alpha_3}^G)$.

$$\text{"}DE/rand/2\text{"} : Y_i^G = X_{\alpha_1}^G + F * (X_{\alpha_2}^G - X_{\alpha_3}^G) + F * (X_{\alpha_4}^G - X_{\alpha_5}^G), \tag{3}$$

where, $rand/2$ shows the second mutation strategy. Here, the base vector $(X_{\alpha_1}^G)$ with addition of two different, difference vectors $[(X_{\alpha_2}^G - X_{\alpha_3}^G), (X_{\alpha_4}^G - X_{\alpha_5}^G)]$.

$$\text{"}DE/best/1\text{"} : Y_i^G = X_{i,best}^G + F * (X_{\alpha_1}^G - X_{\alpha_2}^G). \tag{4}$$

In this mutation strategy, $best/1$ is the best candidate solution of the current generation $G$. $X_{i,best}^G$ is the best individual vector with the best fitness solution.

where $i = [1, 2, \ldots, N_\rho]$, $\alpha_1, \alpha_2, \alpha_3, \alpha_4, \alpha_5$ are randomly chosen integers from $N_\rho$ such that $[\alpha_1 \neq \alpha_2 \neq \alpha_3 \neq \alpha_4 \neq \alpha_5] \neq i$, it means that these randomly chosen vector should not be same for the particular iterations along with corresponding. In-order to get the fined results the mutation scaling factor should be under $F \in [0, 1]$ ranges. In the present study, the basic mutation strategy (Eq. (2)) is used for mutation.

## Crossover

Previous studies indicate that crossover functions as a filter for the population derived from the initial population $X_i^G = (x_{1,i}^G, x_{2,i}^G, \ldots, x_{D,i}^G)$ and the mutated population $Y_i^G = (y_{1,i}^G, y_{2,i}^G, \ldots, y_{D,i}^G)$, ultimately producing a distinct trial population $Z_i^G = (z_{1,i}^G, z_{2,i}^G, \ldots, z_{D,i}^G)$. The proposed eMDE incorporates the binomial crossover, defined as follows:

$$z_{j,i}^G = \begin{cases} y_{j,i}^G & \text{if} \quad rand_j \leq Cr \text{ or } j = j_{rand}, \\ x_{j,i}^G & \text{otherwise}, \quad j = 1, 2, 3, \ldots, D, \end{cases} \tag{5}$$

where, $j_{rand}$ is a random index chosen from $j = 1, 2, 3, \ldots, D$ for a particular vector of population $Y_i^G$ and $rand_j \varepsilon [0, 1]$.

## Selection strategy

The technique primarily facilitates the production of the subsequent population for the following generation (G+1), derived from the correspondence between the present population $(X_i^G)$ and the trial population $(Z_i^G)$. This is the tournament selection method employed for the selection process.

$$X_i^{G+1} = \begin{cases} Z_i^G & \text{if} \quad (Z_i^G) \leq f(X_i^G) \\ X_i^G & \text{otherwise.} \end{cases} \tag{6}$$

In this selection method, $f(Z_i^G)$ and $f(X_i^G)$ represent the respective fitness values of the trial population $(Z_i^G)$ and the target population $(X_i^G)$.

## PROPOSED DIFFERENTIAL EVOLUTION ALGORITHM (EMDE)

In this section, the proposed DE has been introduced. Basically, the working and involved major changes have been discussed in this particular section. Numpy library of python platform has been used to generate the base population with $N\rho$ number of candidate solutions having D dimension. An enhanced mutation strategy has been adapted to make the variant more efficient and effective in handling large areas of problem functions.

## Novel mutation strategy

In order to reach optimality, there are several algorithms already existing in the studies. But it still remains a big challenge to get the optimum solution. From the literature, it can be concluded that the mutation strategy directly affects the performance of DE algorithm. It measures over two important criteria, convergence rate and quality of the candidate solutions. Therefore, the proposed DE has specially focused on this particular phase with the desire of achieving both criteria. The basic mutation strategy (Eq. (2)) has been adapted and an enhanced mutation strategy has been proposed in order to get a better solution to a vast range of problems.

According to the study, the randomly selected base vector is responsible for the convergence rate of DE and it bears the responsibility of converging the population with the respective generation. So, we have decided to enhance the exploitation of the base vector in Eq. (2). The main motive behind this proposed approach, where the base vector is multiplied with the co-efficient factor "$\sigma$", is that the coefficient factor "$\sigma$" is computed with the logarithmic inspired formula given in Eq. (8) and is measured for corresponding generation 'G' upto maximum generation $G_{max}$ which basically increases the ability and quality of convergence rate. It provides the possibility of more finest exploitation for local optimal candidate solutions and extensively explores the chance of reaching global optimality. eMDE has provided a very exclusive impact on population diversity with this newly developed mutation strategy. It has effectively boosted the over-all performance of proposed DE, while comparing with other variants of DE over the many fitness functions.

$$Y_i^G = \sigma_i \cdot X_{\alpha_1}^G + F \cdot (X_{\alpha_2}^G - X_{\alpha_3}^G) \tag{7}$$

where $\sigma_i^G = [\log(e)]^{(G/G_{\max})}$. (8)

In this approach, $\sigma_i$ dynamically adjusts itself based on the generation G, allowing for a adaptive mutation operation. You can experiment with different forms of adaptive scaling or incorporate additional adaptive strategies for the control parameters (F and CR) to further enhance the algorithm's convergence.

### Pseudo code of proposed algorithm

The Algorithm 1 demonstrates the procedures of proposed strategies of the enhanced DE.

## STANDARD BENCHMARK/FITNESS FUNCTION

Although, there are several variants of DE algorithm, It is hard to find-out, which one is best of among algorithms. So, there are many standard mathematical fitness functions involved to measure the performance and accuracy of these algorithm. In the present study, 27 benchmark functions (*Tanabe & Fukunaga, 2013*) have been used (shown in Table 1) to measure the performance of proposed DE along with comparative analysis to other standard DE. As shown in Table 1, $f1 - f11$ are uni-model functions, $f12$ is a discontinuous function, and $f13$–$f27$ are multi-model functions. Each function is defined with boundary limits to generate the population structure.

### Experimental and parameter settings

The control parameters $(F)$ and $(C_r)$ are fixed with the values of 0.5 and 0.8, respectively. $N\rho$ (defined as 100) represents the number of candidate solution. The results of eMDE are compared (listed in Table 2) with the state-of-the-arts DE like jDE (*Brest et al., 2006*), JADE (*Zhang & Sanderson, 2009*), SaDE (*Qin, Huang & Suganthan, 2009*), CoDE (*Wang, Cai & Zhang, 2011*), EPSDE (*Mallipeddi et al., 2011*), MPEDE (*Wu et al., 2016*), and DISDE (*Zhong & Cheng, 2020*) over 27 benchmark functions bearing 30 and 100 dimensions. Further, in Table 3, eMDE is also compared over 10 benchmark functions listed in Table 1 with respect to simple DE, CPDE and CPMDE (*Pooja, Chaturvedi & Kumar, 2015*) over 30 and 50 dimensions, respectively. Furthermore, in Table 4, The performance of proposed DE (eMDE) have been tested over 11 benchmark functions from Table 1 with swarm intelligence based Whale Optimization Algorithm (WOA) (*Mirjalili & Lewis, 2016*) over 50 and 100 dimensions respectively.

The performance of the proposed DE (eMDE) have been tested and the result has been compared in Tables 2–4. All the benchmark functions have been tested and evolved for 50 generations and each of the generations has been processed for 200,000 iterations.

The testing findings were acquired using a Windows 10 operating system, an Intel i5 processor, and an 8 GB RAM configuration. The proposed technique is developed using the Python programming language on the Python 3.0 platform.

## COMPARATIVE STUDY AND DISCUSSION

The results of eMDE are compared with the state-of-the-arts of DE like jDE (*Brest et al., 2006*), JADE (*Zhang & Sanderson, 2009*), SaDE (*Qin, Huang & Suganthan, 2009*), CoDE (*Wang, Cai & Zhang, 2011*), EPSDE (*Mallipeddi et al., 2011*), MPEDE (*Wu et al., 2016*),

**Algorithm 1 Pseudo code of proposed algorithm.**

**Initialize Population**

$N\rho$, number of candidate solutions,

$D$, dimension of a vector,

**for** $run \leftarrow 0$ $to$ $max\_run$ **do**

  **for** $iter \leftarrow 0$ $to$ $max\_iter$ **do**

    $pop \leftarrow \text{scale}(pop, \{LowBound, HighBound\})$

    $mutate\_pop, trial\_pop, selection \leftarrow \text{list}(), \text{list}(), \text{list}()$

    **for** $i \leftarrow 0$ $to$ $pop\_size$ **do**

      $\alpha_1, \alpha_2, \alpha_3 \leftarrow \text{init}_p\text{op}[\text{random}_i\text{ndex}], \text{init}_p\text{op}[\text{random}_i\text{ndex}], \text{init}_p\text{op}[\text{random}_i\text{ndex}]$

      $\sigma_i^G \leftarrow [\log(e)]^{(G/G_{\max})}$

      $novel\_mutation \leftarrow \sigma_i^G \cdot \alpha_1 + F \cdot (\alpha_2 - \alpha_3 + novel\_scale \cdot \text{rand}())$

      $mutate\_pop.\text{append}(novel\_mutation)$

    **end for**

    **for** $j \leftarrow 0$ $to$ $pop\_size$ **do**

      $Threshold(CR) \leftarrow 0.8$

      $cross\_points \leftarrow \text{random}(0,1) < CR$

      **if** $\neg\text{any}(cross\_points)$ **then**

        $cross\_points[\text{random}(0,D)] \leftarrow \text{True}$

      **end if**

      $trial\_pop.\text{append}(\text{where}(cross\_points, mutate\_pop[j], \text{init}_p\text{op}[j]))$

    **end for**

    **for** $k \leftarrow 0$ $to$ $pop\_size$ **do**

      $score\_trial, score\_target \leftarrow \text{benchmark}_f\text{n}(trial\_pop[k]), \text{benchmark}_f\text{n}(\text{init}_p\text{op}[k])$

      **if** $score\_trial < score\_target$ **then**

        $\text{init}_p\text{op}[k] \leftarrow \text{trial}_p\text{op}[k]$

        $gen\_scores.\text{append}(score\_trial)$

      **else**

        $gen\_scores.\text{append}(score\_target)$

      **end if**

    **end for**

  **end for**

**end for**

and DISDE (*Zhong & Cheng, 2020*) over 27 benchmark functions bearing 30 and 100 D and presented in Table 2. Further, in Table 3, the proposed DE (eMDE) is compared over 10 benchmark functions with respect to simple DE, CPDE and CPMDE (*Pooja, Chaturvedi & Kumar, 2015*) for 30 and 50 D respectively. Later, the performance of proposed DE (eMDE) has also been validated with swarm intelligence based Whale

**Table 1 Standard benchmark function.**

| Func | Function name | Bounds [Low, Up] | Global_optima, f(x*) |
|---|---|---|---|
| f1 | Sphere | [−100, 100] | 0 |
| f2 | Elliptic | [−100, 100] | 0 |
| f3 | Bent cigar | [−100, 100] | 0 |
| f4 | Schwefel 1.2 | [−100, 100] | 0 |
| f5 | Schwefel 2.22 | [−10, 10] | 0 |
| f6 | Schwefel 2.21 | [−100, 100] | 0 |
| f7 | Powell sum | [−100, 100] | 0 |
| f8 | Sum square | [−10, 10] | 0 |
| f9 | Discuss | [−100, 100] | 0 |
| f10 | Different powers | [−100, 100] | 0 |
| f11 | Zakharov | [−5, 10] | 0 |
| f12 | Step | [−100, 100] | 0 |
| f13 | Noise quartic | [−1.28, 1.28] | 0 |
| f14 | Rosenbrock | [−30, 30] | 0 |
| f15 | Griewank | [−600, 600] | 0 |
| f16 | Rastrigin | [−5.12, 5.12] | 0 |
| f17 | Apline | [−100, 100] | 0 |
| f18 | Bohachevsky_2 | [−100, 100] | 0 |
| f19 | Salomon | [−100, 100] | 0 |
| f20 | Scaffer2 | [−100, 100] | 0 |
| f21 | Weierstrass | [−0.5, 0.5] | 0 |
| f22 | Katsuura | [−100, 100] | 0 |
| f23 | HappyCat | [−100, 100] | 0 |
| f24 | HGBat | [−100, 100] | 0 |
| f25 | Scaffer's f6 | [−0.5, 0.5] | 0 |
| f26 | Expanded Scaffer's f6 | [−5, 5] | 0 |
| f27 | Expanded Griewank's plus Rosenbrock's | [−5.12, 5.12] | 0 |

**Table 2 Comparative study of eMDE with state-of-the-arts in terms of mean fitness and standard deviation.**

| Function name | D | Fitness value (mean/sdv) | | | | | | | |
|---|---|---|---|---|---|---|---|---|---|
| | | jDE | JADE | SaDE | CoDE | EPSDE | MPEDE | DSIDE | eMDE |
| f1 | 30 | 8.7e−18/7.6e−18 | 4.6e−36/2.2e−35 | 5.2e−24/4.9e−24 | 5.9e−08/2.5e−08 | 1.4e−26/4.4e−26 | 1.5e−29/7.9e−29 | 0.0e+00/0.0e+00 | 0.0e+00/0.0e+00 |
| | 100 | 1.7e−05/7.3e−06 | 3.0e−16/3.2e−16 | 2.9e−04/1.3e−04 | 7.4e−01/2.7e−01 | 2.1e−08/2.2e−08 | 9.2e−10/6.7e−10 | 0.00e+00/0.00e+00 | 0.0e+00/0.0e+00 |
| f2 | 30 | 2.1e−14/1.6e−14 | 2.8e−31/1.2e−30 | 1.0e−20/1.6e−20 | 4.1e−05/1.7e−05 | 2.3e−22/3.9e−22 | 3.8e−27/1.7e−26 | 0.0e+00/0.0e+00 | 0.0e+00/0.0e+00 |
| | 100 | 8.4e−02/5.8e−02 | 4.1e−11/4.2e−11 | 9.0e−01/5.3e−01 | 2.1e+03/6.1e+02 | 2.4e−04/2.9e−04 | 1.4e−04/1.8e−04 | 0.0e+00/0.0e+00 | 0.0e+00/0.0e+00 |
| f3 | 30 | 6.6e−12/4.9e−12 | 1.0e−30/2.9e−30 | 3.2e−18/2.9e−18 | 3.9e−02/1.4e−02 | 3.7e−21/9.0e−21 | 1.8e−25/6.4e−25 | 0.0e+00/0.0e+00 | 0.0e+00/0.0e+00 |
| | 100 | 1.6e+01/7.8e+00 | 1.2e−09/1.7e−09 | 3.4e+02/2.2e+02 | 7.2e+05/1.9e+05 | 2.0e−02/1.2e−02 | 5.1e−13/1.2e−04 | 0.0e+00/0.0e+00 | 0.0e+00/0.0e+00 |
| f4 | 30 | 4.1e+00/5.4e+00 | 1.2e−13/1.4e−13 | 5.9e−01/4.6e−01 | 6.9e−02/3.7e−02 | 7.8e−01/4.4e+00 | 1.3e−06/7.2e−06 | 0.0e+00/0.0e+00 | 0.0e+00/0.0e+00 |
| | 100 | 8.1e+03/2.9e+03 | 2.3e+02/1.0e+02 | 1.6e+03/3.0e+02 | 1.4e+03/3.2e+02 | 2.9e+04/3.0e+04 | 2.9e+02/1.3e+02 | 0.0e+00/0.0e+00 | 0.0e+00/0.0e+00 |
| f5 | 30 | 3.4e−11/1.3e−11 | 1.1e−14/3.2e−14 | 7.0e−14/3.1e−14 | 5.0e−05/9.5e−06 | 8.7e−11/1.2e−10 | 7.7e−16/1.9e−15 | 0.0e+00/0.0e+00 | 0.0e+00/0.0e+00 |
| | 100 | 7.7e−04/3.3e−04 | 3.2e−08/3.1e−08 | 1.2e−03/1.8e−03 | 1.1e+00/2.0e−01 | 8.0e−05/1.1e−04 | 3.9e−03/1.9e−02 | 0.0e+00/0.0e+00 | 0.0e+00/0.0e+00 |

| Function name | D | Fitness value (mean/sdv) | | | | | | | |
|---|---|---|---|---|---|---|---|---|---|
| | | jDE | JADE | SaDE | CoDE | EPSDE | MPEDE | DSIDE | eMDE |
| f6 | 30 | 8.4e−01/8.2e−01 | 8.9e−12/1.8e−11 | 4.6e−02/7.7e−02 | 3.2e−01/8.0e−02 | 1.4e−01/4.4e−01 | 1.6e−07/1.8e−07 | 0.0e+00/0.0e+00 | 0.0e+00/0.0e+00 |
| | 100 | 2.8e+01/4.3e+00 | 7.1e+00/9.7e−01 | 1.1e+01/1.9e+00 | 5.0e+00/7.3e−01 | 6.8e+01/1.9e+01 | 1.0e+01/1.6e+00 | 0.0e+00/0.0e+00 | 3.2e−03/1.2e−02 |
| f7 | 30 | 3.1e−41/1.6e−40 | 9.3e−44/4.2e−43 | 4.4e−22/1.9e−21 | 2.1e−17/5.2e−17 | 3.9e−38/2.0e−28 | 6.1e−53/3.2e−52 | 0.0e+00/0.0e+00 | 0.0e+00/0.0e+00 |
| | 100 | 3.6e+27/1.8e+28 | 5.2e+33/2.7e+34 | 9.4e+53/5.1e+54 | 8.3e+20/3.1e+21 | 1.0e+41/5.3e+41 | 1.4e+38/7.6e+38 | 0.0e+00/0.0e+00 | 0.0e+00/0.0e+00 |
| f8 | 30 | 1.3e−18/1.3e−18 | 2.9e−37/1.0e−36 | 6.1e−25/5.9e−25 | 6.7e−09/2.7e−09 | 3.3e−28/5.3e−28 | 4.9e−32/2.7e−31 | 0.0e+00/0.0e+00 | 0.0e+00/0.0e+00 |
| | 100 | 7.8e−06/3.1e−06 | 2.0e−16/2.8e−61 | 1.1e−04/5.9e−05 | 2.9e−01/6.8e−02 | 6.5e−09/3.4e−09 | 8.2e−10/6.9e−10 | 0.0e+00/0.0e+00 | 0.0e+00/0.0e+00 |
| f9 | 30 | 1.8e−17/2.0e−17 | 1.4e−32/7.8e−32 | 1.2e−23/1.0e−23 | 8.2e−08/3.3e−08 | 1.2e−25/2.7e−25 | 9.10e−2/4.6e−28 | 0.0e+00/0.0e+00 | 0.0e+00/0.0e+00 |
| | 100 | 2.9e−05/1.2e−05 | 9.4e−16/7.4e−16 | 5.4e−04/2.9e−04 | 9.3e−01/2.9e−01 | 2.0e−07/1.0e−07 | 6.0e−09/7.7e−09 | 0.0e+00/0.0e+00 | 0.0e+00/0.0e+00 |
| f10 | 30 | 9.1e−13/5.9e−13 | 1.0e−23/3.3e−23 | 1.9e−12/2.8e−12 | 5.1e−06/1.7e−06 | 1.3e−18/1.9e−18 | 1.1e−02/3.9e−02 | 0.0e+00/0.0e+00 | 0.0e+00/0.0e+00 |
| | 100 | 1.9e−02/3.3e−02 | 2.6e−08/1.9e−08 | 2.9e−01/9.0e−02 | 1.4e+00/4.0e−01 | 3.4e−03/2.9e−03 | 1.2e−05/7.3e−06 | 0.0e+00/0.0e+00 | 0.0e+00/0.0e+00 |
| f11 | 30 | 2.2e−15/3.0e−15 | 1.2e−39/6.8e−39 | 1.1e−21/1.1e−21 | 2.0e−08/9.1e−09 | 1.4e−26/5.6e−26 | 7.9e−34/2.0e−33 | 0.0e+00/0.0e+00 | 0.0e+00/0.0e+00 |
| | 100 | 1.2e−03/7.0e−04 | 8.3e−16/6.3e−16 | 2.2e−04/1.2e−04 | 1.9e−02/5.9e−03 | 7.9e−06/8.2e−06 | 2.7e−09/2.9e−09 | 0.0e+00/0.0e+00 | 0.0e+00/0.0e+00 |
| f12 | 30 | 9.6e−18/1.0e−17 | 3.0e−34/1.2e−33 | 4.9e−24/6.2e−24 | 6.1e−08/1.4e−08 | 5.9e−27/1.8e−26 | 2.1e−31/6.9e−31 | 1.1e+00/2.0e−01 | 0.0e+00/0.0e+00 |
| | 100 | 1.9e−05/9.7e−06 | 2.6e−16/2.3e−16 | 2.9e−04/1.3e−04 | 7.9e−01/2.3e−01 | 1.6e−08/8.6e−09 | 9.6e−10/1.0e−09 | 1.1e+01/5.9e+00 | 1.3e−32/2.4e−32 |
| f13 | 30 | 1.1e−02/3.2e−03 | 3.9e−03/1.6e−03 | 4.9e−03/1.9e−03 | 1.3e−02/3.8e−03 | 4.8e−03/2.2e−03 | 3.4e−03/1.7e−03 | 8.4e−04/7.6e−04 | 3.9e−04/2.6e−04 |
| | 100 | 7.1e−02/1.2e−02 | 4.2e−02/9.9e−03 | 1.3e−01/2.9e−02 | 6.4e−02/1.8e−02 | 5.1e−02/2.2e−02 | 7.1e−02/1.9e−02 | 9.7e−04/1.1e−03 | 2.8e−01/1.4e−01 |
| f14 | 30 | 2.7e+01/1.3e+01 | 1.7e−01/8.3e−01 | 2.9e+01/1.4e+01 | 2.2e+01/5.3e−01 | 1.0e+01/3.2e+00 | 3.1e+00/4.0e+00 | 2.9e+01/9.9e−02 | 0.0e+00/0.0e+00 |
| | 100 | 1.9e+02/5.8e+01 | 1.4e+02/4.9e+01 | 4.7e+02/8.0e+01 | 2.0e+02/3.9e+01 | 1.9e+02/5.1e+01 | 1.8e+02/6.4e+01 | 9.9e+01/7.3e−02 | 2.7e−28/2.2e−27 |
| f15 | 30 | 0.0e+00/0.0e+00 | 1.8e−11/9.8e−11 | 1.7e−03/4.2e−03 | 1.9e−05/4.4e−05 | 0.0e+00/0.0e+00 | 1.4e−03/3.8e−03 | 0.0e+00/0.0e+00 | 0.0e+00/0.0e+00 |
| | 100 | 9.7e−06/4.0e−06 | 2.4e−03/7.9e−03 | 7.8e−03/1.2e−02 | 4.1e−01/1.2e−01 | 1.3e−03/3.6e−03 | 4.9e−03/9.7e−03 | 0.0e+00/0.0e+00 | 0.0e+00/0.0e+00 |
| f16 | 30 | 1.4e−04/7.0e−04 | 1.9e−04/8.9e−05 | 5.3e−01/8.1e−01 | 2.4e+01/1.9e+00 | 6.2e−01/7.9e−01 | 2.8e−12/6.0e−12 | 0.0e+00/0.0e+00 | 2.6e−02/3.2e−02 |
| | 100 | 1.9e+03/1.6e+01 | 1.6e+02/9.9e+00 | 2.7e+03/1.3e+01 | 7.1e+02/2.1e+01 | 4.3+02/2.7e+01 | 4.3e+01/8.9e+00 | 0.0e+00/0.0e+00 | 3.2e+01/2.6e+01 |
| f17 | 30 | 1.4e−03/4.8e−04 | 1.1e−02/3.9e−03 | 8.2e−04/3.8e−04 | 3.0e+01/3.9e+00 | 1.4e−02/5.2e−03 | 2.2e−07/1.2e−06 | 0.0e+00/0.0e+00 | 0.0e+00/0.0e+00 |
| | 100 | 1.9e+00/2.2e+00 | 1.6e+01/5.4e+00 | 3.0e+00/4.1e+00 | 8.9e+01/1.2e+01 | 1.7e+01/2.63e+01 | 2.4e+00/2.3e+00 | 0.0e+00/0.0e+00 | 0.0e+00/0.0e+00 |
| f18 | 30 | 1.3e−16/2.1e−16 | 0.0e+00/0.0e+00 | 9.3e−02/3.0e−01 | 4.3e−06/2.0e−06 | 0.0e+00/0.0e+00 | 3.9e−02/2.1e−01 | 0.0e+00/0.0e+00 | 0.0e+00/0.0e+00 |
| | 100 | 6.15e−0/3.0e−04 | 2.3e+00/1.4e+00 | 7.0e+00/2.8+00 | 2.0e+01/3.7e+00 | 2.4e+00/2.0e+00 | 5.0e+00/2.4e+00 | 0.00e+00/0.0e+00 | 0.0e+00/0.0e+00 |
| f19 | 30 | 2.1e−01/3.4e−02 | 2.0e−01/1.9e−02 | 2.1e−01/3.4e−02 | 3.7e−01/4.7e−02 | 1.7e−01/4.7e−02 | 2.6e−01/5.7e−02 | 0.0e+00/0.0e+00 | 9.9e−02/7.9e−02 |
| | 100 | 9.9e−01/8.8e−02 | 7.0e−01/1.0e−01 | 1.7e+00/2.2e−01 | 1.9e+00/1.3e−01 | 1.0e+00/1.9e−01 | 3.3e+00/2.9e−01 | 0.0e+00/0.0e+00 | 1.0e−01/2.3e−02 |
| f20 | 30 | 5.7e+00/1.3e+00 | 4.7e+00/7.9e−01 | 1.7e2+01/2.7e+00 | 2.0e+01/2.9e+00 | 1.1e−02/1.7+00 | 2.2e+00/1.1e+00 | 0.0e+00/0.0e+00 | 0.0e+00/0.0e+00 |
| | 100 | 9.0e+01/9.5e+00 | 8.9e+01/6.9e+00 | 1.6e+02/2.0e+01 | 3.0e+02/9.1e+00 | 2.9e+02/2.6e+01 | 1.2e+01/4.9e+00 | 0.0e+00/0.0e+00 | 0.0e+00/0.0e+00 |
| f21 | 30 | 3.3e−10/3.9e−10 | 2.4e−05/3.9e−05 | 7.4e−14/8.7e−14 | 1.2e−02/1.6e−03 | 6.4e+00/2.3e+00 | 4.3e−03/1.4e−02 | 0.0e+00/0.0e+00 | 3.2e−01/4.2e−01 |
| | 100 | 5.8e−02/1.6e−02 | 2.8e+00/1.4e+00 | 2.9e+00/1.3e+00 | 6.0e+00/7.8e−01 | 1.0e+02/2.9e+00 | 1.3e+01/2.9e+00 | 0.0e+00/0.0e+00 | 6.4e+00/5.8e+00 |
| f22 | 30 | 5.7e−02/8.3e−03 | 4.7e+00/7.9e−01 | 1.4e−01/1.9e−02 | 8.9e−03/7.0e−04 | 8.9e−02/1.3e−02 | 6.9e−04/1.4e−04 | 0.0e+00/0.0e+00 | 0.0e+00/0.0e+00 |
| | 100 | 2.2e−01/2.6e−02 | 1.0e−01/9.1e−03 | 5.4e−01/3.9e−02 | 4.0e−01/3.5e−02 | 5.9e−01/3.8e−02 | 9.3e−03/1.7e−03 | 0.0e+00/0.0e+00 | 0.0e+00/0.0e+00 |
| f23 | 30 | 3.7e−01/5.2e−02 | 2.6e−01/3.7e−02 | 3.6e−01/6.1e−02 | 4.9e−01/4.8e−02 | 3.3e−01/3.9e−02 | 2.8e−01/5.3e−02 | 5.0e−01/6.3e−02 | 3.3e−05/4.6e−05 |
| | 100 | 6.3e−01/5.3e−02 | 5.1e−01/6.2e−02 | 6.2e−01/7.9e−02 | 7.2e−01/8.3e−02 | 5.9e−01/5.9e−02 | 5.8e−01/7.3e−02 | 8.8e−01/4.6e−02 | 5.8e−04/3.2e−03 |
| f24 | 30 | 3.3e−01/3.8e−02 | 3.8e−01/1.2e−01 | 4.0e−01/1.1e−01 | 2.9e−01/2.8e−02 | 3.4e−01/7.4e−02 | 4.1e−01/1.7e−01 | 4.9e+01/1.6e−02 | 1.0e−02/2.6e−01 |
| | 100 | 5.0e−01/1.8e−01 | 5.0e−02/2.0e−01 | 6.1e−01/1.9e−01 | 6.3e−01/2.3e−01 | 5.8e−01/1.9e−01 | 5.4e−01/2.3e−01 | 4.9e−01/7.9e−04 | 4.2e−02/3.4e−02 |
| f25 | 30 | 0.0e+00/0.0e+00 | 0.0e+00/0.0e+00 | 0.0e+00/0.0e+00 | 2.4e−12/1.1e−12 | 0.0e+00/0.0e+00 | 0.0e+00/0.0e+00 | 0.0e+00/0.0e+00 | 0.0e+00/0.0e+00 |
| | 100 | 1.0e−09/7.4e−01 | 1.4e−16/3.1e−16 | 1.6e−08/8.7e−09 | 3.6e−05/1.1e−05 | 9.6e−13/6.1e−03 | 1.0e−13/1.9e−13 | 0.0e+00/0.0e+00 | 0.0e+00/0.0e+00 |
| f26 | 30 | 7.9e−01/1.3e−01 | 6.8e−01/6.1e−02 | 1.2e+00/5.7e−01 | 1.8e+00/1.8e−01 | 2.0e+00/2.4e−01 | 3.4e−01/7.0e−02 | 0.0e+00/0.0e+00 | 0.0e+00/0.0e+00 |
| | 100 | 1.0e+01/8.8e−01 | 9.0e+00/3.9e−01 | 1.6e+01/6.6e−01 | 2.2e+01/8.4e−01 | 2.8e+01/1.9e+00 | 2.3e+00/7.0e−01 | 0.0e+00/0.0e+00 | 0.0e+00/0.0e+00 |
| f27 | 30 | 3.2e+00/3.4e−01 | 2.9e+00/2.0e−01 | 6.2e+00/5.7e−01 | 8.4e+00/6.8e−01 | 4.6e+00/4.3e−01 | 2.1e+00/2.6e−01 | 1.1e+01/4.7e−01 | 1.0e+00/1.2e+00 |
| | 100 | 3.3e+01/3.1e+00 | 2.9e+01/1.8e+00 | 4.9e+01/2.0e+00 | 7.0e+01/2.1e+00 | 6.4e+01/7.4e+00 | 1.9e+01/2.9e+00 | 4.4e+01/5.3e−01 | 5.2e+01/4.4e+01 |

**Table 3 Comparative study of eMDE with DE, CPDE and CPMDE in terms of mean fitness and standard deviation.**

| Function name | D | Fitness value (mean) | | | | Fitness value (sdv) | | | | Performance ( +, −, ~) |
|---|---|---|---|---|---|---|---|---|---|---|
| | | DE | CPDE | CPMDE | eMDE | DE | CPDE | CPMDE | eMDE | |
| Sphere | 30 | 6.09141e−32 | 8.8785e−59 | 1.72634e−111 | 0.00e+00 | 4.62959e−32 | 7.34566e−59 | 2.73492−111 | 0.00e+00 | + |
| | 50 | 2.3966e−19 | 6.69419e−38 | 1.80039e−74 | 0.00e+00 | 1.74151e−19 | 6.00656e−38 | 8.51649e−75 | 0.00e+00 | + |
| Schwefel_1.2 | 30 | 3.27407e−05 | 2.48188e−06 | 2.72926e−18 | 0.00e+00 | 1.95386e−05 | 1.96591e−06 | 2.53122e−18 | 0.00e+00 | + |
| | 50 | 8.16825e+01 | 1.4119e+01 | 7.22052e−04 | 0.00e+00 | 3.60447e+01 | 3.60447e−00 | 8.30536e−04 | 0.00e+00 | + |
| Schwefel_2.22 | 30 | 4.43384e−16 | 7.8695e−32 | 6.74805e−60 | 0.00e+00 | 1.99391e−16 | 4.85753e−32 | 4.24994e−60 | 0.00e+00 | + |
| | 50 | 5.64878e−10 | 1.55056e−21 | 5.84604−42 | 0.00e+00 | 1.69532−10 | 9.90536e−22 | 2.31529−42 | 0.00e+00 | + |
| Schwefel_2.21 | 30 | 1.86659e−05 | 0.00e+00 | 0.00e+00 | 0.00e+00 | 0.00e+00 | 0.00e+00 | 0.00e+00 | 0.00e+00 | ~ |
| | 50 | 7.26021e+00 | 0.00e+00 | 0.00e+00 | 1.1853e−04 | 2.15332e+00 | 0.00e+00 | 0.00e+00 | 1.0542e−04 | − |
| Zakharov | 02 | 0.00e+00 | 0.00e+00 | 0.00e+00 | 0.00e+00 | 0.00e+00 | 0.00e+00 | 0.00e+00 | 0.00e+00 | ~ |
| Step | 30 | 0.00e+00 | 0.00e+00 | 0.00e+00 | 0.00e+00 | 0.00e+00 | 0.00e+00 | 0.00e+00 | 0.00e+00 | ~ |
| | 50 | 0.00e+00 | 0.00e+00 | 0.00e+00 | 0.00e+00 | 0.00e+00 | 0.00e+00 | 0.00e+00 | 0.00e+00 | ~ |
| Noise quartic | 30 | 3.13412e−03 | 2.00707e−03 | 1.22697e−03 | 3.976e−05 | 4.23781e−04 | 3.74599e−04 | 1.61742e−04 | 2.5854e−05 | + |
| | 50 | 1.06329e−02 | 8.30663e−03 | 4.9343e−03 | 1.846e−04 | 2.09914e−03 | 1.88658e−03 | 1.34447e−03 | 2.8967e−04 | + |
| Rosenbrock | 30 | 1.01013e+00 | 2.4619e+01 | 1.07474e+01 | 0.00e+00 | 1.17281e+00 | 1.0052e+00 | 9.78158e+00 | 0.00e+00 | + |
| | 50 | 3.28565e+01 | 8.0669e+01 | 5.36856e+01 | 0.00e+00 | 8.3860e−01 | 8.0669e+01 | 2.09962+01 | 0.00e+00 | + |
| Griewank | 30 | 0.00e+00 | 0.00e+00 | 0.00e+00 | 0.00e+00 | 0.00e+00 | 0.00e+00 | 0.00e+00 | 0.00e+00 | ~ |
| | 50 | 0.00e+00 | 0.00e+00 | 0.00e+00 | 0.00e+00 | 0.00e+00 | 0.00e+00 | 0.00e+00 | 0.00e+00 | ~ |
| Rastrigin | 15 | 2.66469+01 | 0.00e+00 | 3.23362e+00 | 0.00e+00 | 6.65224e+00 | 0.00e+00 | 4.3083e−01 | 0.00e+00 | ~ |

Optimization Algorithm (WOA) (*Mirjalili & Lewis, 2016*) over 50 and 100D respectively for 11 benchmark functions and results are contained in Table 4.

## Comparison of eMDE with state-of-the-arts algorithms

In Table 2, the results of eMDE are shown and compared to some state-of-the-art in terms of mean fitness and its standard deviation. From the results, it is quite clear that eMDE outperforms four out of 27 functions than all other variants of DE over 30 and 100D respectively. However, for function $f13$ and $f27$ with 30D and $f6$ with 100D, eMDE produces better results than all other competitive DE variants.

For function $f1$–$f5$, $f7$–$f11$, $f15$, $f17$–$f20$, $f22$, $f25$ and $f26$, eMDE is performing better or similar with the other listed DE algorithm over 30 and 100D, respectively. Moreover for function $f16$, over 30D, the proposed DE is performing better then SaDE, EPSDE, and CoDE. Further, for $f16$ and $f19$, it outperforms than jDE, JADE, SaDE, EPSDE, CoDE and MPEDE except DSIDE for 100D.

For function $f27$ over 100 dimensions, eMDE is performing better than CoDE and EPSDE. For function $f21$, it gives better results with only EPSDE over 30D, while for 100D, it shows betterment for EPSDE and MPEDE.

**Table 4 Comparative study of eMDE with Swarm-based Algorithm (WOA) in terms of mean fitness and standard deviation.**

| Function name | D | Fitness value (mean value) | | Fitness value (sdv) | | Performance ( +, −, ~) |
|---|---|---|---|---|---|---|
| | | WOA | eMDE | WOA | eMDE | |
| Sphere | 50 | 0.00e+00 | 0.00e+00 | 0.00e+00 | 0.00e+00 | ~ |
| | 100 | 0.00e+00 | 0.00e+00 | 0.00e+00 | 0.00e+00 | ~ |
| Schwefel_1.2 | 50 | 0.00e+00 | 0.00e+00 | 0.00e+00 | 0.00e+00 | ~ |
| | 100 | 0.00e+00 | 0.00e+00 | 0.00e+00 | 0.00e+00 | ~ |
| Schwefel_2.22 | 50 | 0.00e+00 | 0.00e+00 | 0.00e+00 | 0.00e+00 | ~ |
| | 100 | 2.2381e−04 | 0.00e+00 | 3.2153e−04 | 0.00e+00 | + |
| Schwefel_2.21 | 50 | 0.00e+00 | 2.0542e−04 | 0.00e+00 | 1.0542e−03 | ~ |
| | 100 | 7.4215e−05 | 3.8941e−05 | 6.5381e−06 | 3.1242e−05 | ~ |
| Zakharov | 50 | 0.00e+00 | 0.00e+00 | 0.00e+00 | 0.00e+00 | ~ |
| | 100 | 0.00e+00 | 0.00e+00 | 0.00e+00 | 0.00e+00 | ~ |
| Step | 50 | 1.4325e−04 | 0.00e+00 | 2.3692e−04 | 0.00e+00 | + |
| | 100 | 8.0947e−02 | 0.00e+00 | 6.2154e−03 | 0.00e+00 | + |
| Noise quartic | 50 | 9.7192e−01 | 1.7561e−05 | 8.3687e−01 | 2.5434e−05 | + |
| | 100 | 3.6571e−01 | 3.1245e−06 | 5.5231e−01 | 9.8941e−05 | + |
| Rosenbrock | 50 | 2.2891e−03 | 0.00e+00 | 2.2545e−03 | 0.00e+00 | + |
| | 100 | 1.2861e−03 | 0.00e+00 | 1.0164e−03 | 0.00e+00 | + |
| Greiwank | 50 | 0.00e+00 | 0.00e+00 | 0.00e+00 | 0.00e+00 | ~ |
| | 100 | 0.00e+00 | 0.00e+00 | 0.00e+00 | 0.00e+00 | ~ |
| Rastringin | 50 | 0.00e+00 | 0.00e+00 | 0.00e+00 | 0.00e+00 | ~ |
| | 100 | 0.00e+00 | 0.00e+00 | 0.00e+00 | 0.00e+00 | ~ |
| Elliptical | 50 | 0.00e+00 | 0.00e+00 | 0.00e+00 | 0.00e+00 | ~ |
| | 100 | 5.1979e−03 | 0.00e+00 | 6.1979e−03 | 0.00e+00 | + |

## Comparison with DE, CPDE and CPMDE

In Table 3, the performance of the proposed DE has been measured against three other variants of DE *i.e.* simple DE, CPDE, and CPMDE over the listed benchmark function for 30D and 50D respectively. It can be clearly visibilized that the proposed DE is superior or competative to all other DE variants for most of the functions. But particularly for Schwefel_2.21 with 50D, eMDE underperformed than CPDE and CPMDE. However, it shows better results than simple DE. The performance evaluation of eMDE has also been shown with some mathematical expression $(+, -, \sim)$ representing the better, worse and equivalent efficiencies of proposed algorithm over the other algorithms. With these expressions, it became very easy to extract the efficiency of the algorithm in comparison to other variants taken under consideration.

## Comparison of eMDE with whale optimization algorithm

In Table 4, eMDE shows competence with the swarm-based WOA. It can be seen that eMDE outperforms for functions Schwefel 2.22 (100D), Step (50D, 100D), Noise quartic

**Table 5 Comparative study of eMDE with recently developed variants of Differential Evolution in terms of mean fitness and standard deviation.**

| Function name | D | Fitness value (mean value) | | | | Fitness value (sdv) | | | Performance ( +, −, ∼) |
|---|---|---|---|---|---|---|---|---|---|
| | | ADE | TCDE | mDE | eMDE | ADE | TCDE | eMDE | |
| Sphere | 50 | 0.00e+00 | 0.00e+00 | 0.00e+00 | 0.00e+00 | 0.00e+00 | 0.00e+00 | 0.00e+00 | ∼ |
| | 100 | 0.00e+00 | 0.00e+00 | 0.00e+00 | 0.00e+00 | 0.00e+00 | 0.00e+00 | 0.00e+00 | ∼ |
| Schwefel_1.2 | 50 | 0.00e+00 | 0.00e+00 | 0.00e+00 | 0.00e+00 | 0.00e+00 | 0.00e+00 | 0.00e+00 | ∼ |
| | 100 | 0.00e+00 | 0.00e+00 | 0.00e+00 | 0.00e+00 | 0.00e+00 | 0.00e+00 | 0.00e+00 | ∼ |
| Schwefel_2.22 | 50 | 0.00e+00 | 0.00e+00 | 0.00e+00 | 0.00e+00 | 0.00e+00 | 0.00e+00 | 0.00e+00 | ∼ |
| | 100 | 1.52e+00 | 1.54e+00 | 1.05e+00 | 0.00e+00 | 1.00e+00 | 1.04e+00 | 0.00e+00 | + |
| Schwefel_2.21 | 50 | 0.00e+00 | 0.00e+00 | 0.00e+00 | 2.05e-04 | 0.00e+00 | 0.00e+00 | 1.05e-03 | ∼ |
| | 100 | 6.85e+00 | 6.75e+00 | 5.24e+00 | 3.89e-05 | 0.00e+00 | 0.00e+00 | 3.12e-05 | ∼ |
| Zakharov | 50 | 0.00e+00 | 0.00e+00 | 0.00e+00 | 0.00e+00 | 0.00e+00 | 0.00e+00 | 0.00e+00 | ∼ |
| | 100 | 0.00e+00 | 0.00e+00 | 0.00e+00 | 0.00e+00 | 0.00e+00 | 0.00e+00 | 0.00e+00 | ∼ |
| Step | 50 | 1.05e+00 | 0.00e+00 | 0.00e+00 | 0.00e+00 | 0.00e+00 | 0.00e+00 | 0.00e+00 | + |
| | 100 | 2.04e+00 | 3.01e+00 | 2.91e+00 | 0.00e+00 | 2.40e+00 | 1.74e+00 | 0.00e+00 | + |
| Noise quartic | 50 | 7.01e+00 | 6.95e+00 | 5.02e+00 | 1.75e-05 | 0.00e+00 | 0.00e+00 | 2.54e-05 | + |
| | 100 | 3.10e+00 | 2.23e+00 | 1.21e+00 | 3.12e-06 | 0.00e+00 | 0.00e+00 | 9.89e-05 | + |
| Rosenbrock | 50 | 1.14e+00 | 1.19e+00 | 2.24e+00 | 0.00e+00 | 0.00e+00 | 0.00e+00 | 0.00e+00 | + |
| | 100 | 0.00e+00 | 0.20e+00 | 0.00e+00 | 0.00e+00 | 0.00e+00 | 0.00e+00 | 0.00e+00 | + |
| Greiwank | 50 | 0.00e+00 | 0.00e+00 | 0.00e+00 | 0.00e+00 | 0.00e+00 | 0.00e+00 | 0.00e+00 | ∼ |
| | 100 | 0.00e+00 | 0.00e+00 | 0.00e+00 | 0.00e+00 | 0.00e+00 | 0.00e+00 | 0.00e+00 | ∼ |
| Rastringin | 50 | 0.00e+00 | 0.00e+00 | 0.00e+00 | 0.00e+00 | 0.00e+00 | 0.00e+00 | 0.00e+00 | ∼ |
| | 100 | 0.00e+00 | 0.00e+00 | 0.00e+00 | 0.00e+00 | 0.00e+00 | 0.00e+00 | 0.00e+00 | ∼ |
| Elliptical | 50 | 0.00e+00 | 0.00e+00 | 0.00e+00 | 0.00e+00 | 0.00e+00 | 0.00e+00 | 0.00e+00 | ∼ |
| | 100 | 3.05e+00 | 2.59e+00 | 1.32e+00 | 0.00e+00 | 0.00e+00 | 0.00e+00 | 0.00e+00 | + |

(50D, 100D), Rosenbrock (50D, 100D) and Elliptical (100D). Rather than these functions, the WOA algorithm perform quite same as proposed DE.

## Comparative study of eMDE with recent developed variants of differential evolution

In Table 5, eMDE shows the competence with the recently developed variants of differential evolution algorithm. The eMDE outperforms better for every listed fitness functions with other variants.

## Statistical analysis

The statistical analysis of eMDE has been shown with other variants of DE listed in Table 2 to validate the results of eMDE. Two rank-based test methods, Friedman and Kruskal-Wallis test, are involved to evaluate the efficiency of algorithms and their results are enclosed in Table 6. These tests have been carried out over 30 and 100 D respectively for the mean fitness values of the functions taken from Table 2. Finally, it is

**Table 6 Statistical analysis of eMDE for results shown in Table 2.**

| Test | D | jDE | JADE | SaDE | CoDE | EPSDE | MPEDE | DSIDE | eMDE |
|------|----|------|------|------|------|-------|-------|-------|------|
| Friedman (rank) | 30 | 5.25 | 3.12 | 5.42 | 6.73 | 4.92 | 3.53 | 2.75 | 2.26 |
| | 100 | 4.80 | 3.10 | 6.23 | 6.77 | 5.50 | 4.53 | 2.05 | 2.16 |
| Kruskal-Wallis (rank) | 30 | 131.13 | 109.28 | 133.87 | 163.00 | 129.93 | 117.38 | 66.67 | 52.59 |
| | 100 | 132.77 | 120.80 | 146.30 | 165.20 | 137.87 | 128.90 | 51.33 | 51.02 |

quite clear from the test outputs that eMDE generates better results than other variants of DE for each case.

## Convergence graph

The convergence graph (Figs. 1A–1D) have been plotted for 30D while Figs. 1E and 1F are drawn for 2D and 15D respectively and Figs. 1G–1J are shown for 50D for functions listed in Table 3 with respect to mean fitness and number of function evaluation (iteration). The horizontal axis displays the optimal interval results (iterations) from the aforementioned separate runs, while the vertical axis indicates the matching convergence values. eMDE outperforms all other specified variants of DE throughout the given fitness functions, with the exception of the Schwefel function (Fig. 1C), where CPMDE surpasses the proposed DE.

## MECHANICAL ENGINEERING DESIGN PROBLEMS

This section discusses five optimisation challenges in mechanical engineering (*Mishra, Pooja & Tripathi, 2024*).

**CEDP-1:** Pressure vessel design problem: The goal of the pressure vessel problem is to devise a design that minimises fabrication costs while balancing safety factors, ensuring regulatory compliance, and enhancing thermal management to improve performance, reduce weight, and minimise lifecycle costs, all while maintaining safety and reliability.

**CEDP-2:** Three-bar truss design problem: Designing a three-bar truss entails constructing a secure framework utilising three bars interconnected by pins at their termini. The aim is to effectively allocate loads while maintaining stability and integrity. Determine the objective, limitations, and forces. Select an appropriate truss design and specify the geometry. Examine internal forces and choose suitable materials. Guarantee stability and refine the design through iterative optimisation.

**CEDP-3:** Tension/compression spring design problem: This mechanical engineering problem focusses on minimising the spring constraint stress, which encompasses oscillation frequency, minimum deflection, shear stress, and outside diameter.

**CEDP-4:** Welded beam design problem: The objective of this mechanical engineering problem-solving is to build a welded beam that can endure specified loads while optimising its size and material consumption. The design must guarantee the beam's structural integrity, stability, and safety while reducing weight and manufacturing expenses. Material characteristics, weld quality, and cross-sectional shape are essential factors to consider. A

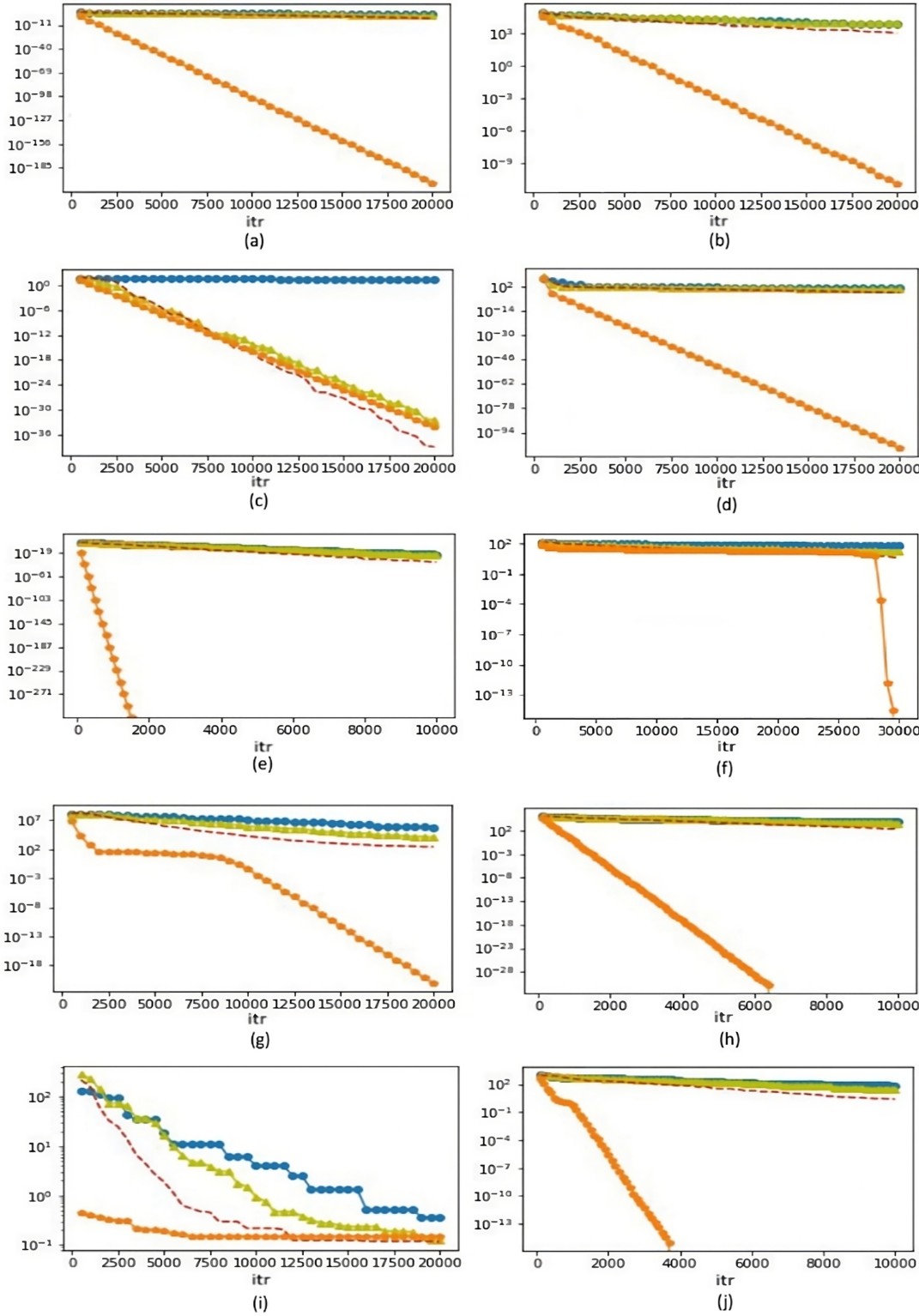

**Figure 1 Convergence graph (eMDE, DE, CPMDE, and CPDE are represented by orange, blue, red, olive-green colour's respectively) for fitness functions listed in Table 3.** (A) Sphere Function (30D), (B) Schwefel Function 1.2 (30D), (C) Schwefel Function 2.21 (30D), (D) Schwefel Function 2.22 (30D), (E) Zakharov Function (02D), (F) Rastrigin Function (15D), (G) Rosenbrock Function (50D), (H) Step Function (50D), (I) Noise Function (50D) and (J) Griewank Function (50D).

comprehensive analysis, encompassing stress and deflection computations, is necessary to ascertain the beam's efficacy under diverse loading scenarios.

**CEDP-5:** Speed reducer design problem: The speed reducer design problem of mechanical engineering involves optimising gear ratios and mechanical components to reduce rotating speed while enhancing torque production. The objective is to attain efficient power transfer, exact control, and reduce energy losses. This crucial activity is essential in numerous applications, including industrial machinery, automotive systems, and robots, guaranteeing seamless and dependable functionality in mechanical systems.

# RESULT ANALYSIS AND DISCUSSION

This section presents the results of the proposed method and compares them with seven other current nature-inspired algorithms.

## Parametric settings

Parametric settings denote the precise configurations or values allocated to parameters inside a model, algorithm, or system. The parametric requirements discussed in the article are presented in the next two sub-subsections.

### Parametric settings for proposed algorithm

The control parameters of differential evolution (DE), specifically the mutation coefficient '$F$' and the crossover rate coefficient '$Cr$', are configured with values of 0.5 and 0.8, respectively. The total candidate solution '$N\rho$' is 100. The parametric settings for constrained engineering design problems are presented in Table 7.

### System requirement for proposed algorithm

The testing results were acquired using a Windows 11 operating system, utilising an Intel i7 processor and 8GB of RAM. The proposed technique is built using the Python programming language on the Python 3.13.0 platform.

## Outcomes

This section provides a concise overview of the results obtained. Eight distinct nature-inspired algorithms were evaluated across five mechanical engineering problems sourced from CEC's 2021 (*Premkumar et al., 2021*). Table 8 demonstrates that the proposed algorithm "eMDE" attained the optimal results regarding mean and standard deviation, that use 2,500 net function evaluations. Additionally, it is compared with seven other nature-inspired algorithms addressing the same problem ("Pressure vessel"). Table 9 presents the "three-bar truss design problem," a mechanical engineering challenge. The proposed algorithm, "eMDE," attained the optimal results (mean and standard deviation) compared to other nature-inspired algorithms, applying 3,500 function evaluations. Table 10 presents the results for the "Tension/compression spring" problem, indicating that the proposed algorithm (eMDE) achieves superior performance (mean and standard deviation) compared to all other algorithms with 3,000 function evaluations. Table 11 demonstrates that the proposed nature-inspired algorithm surpasses the optimal results of all other listed algorithms, achieving 3,200 NFE for the mechanical engineering problem

**Table 7 Parametric settings for constrained engineering design problems.**

| CEDP problems | Dimenssion | Linear inequalities | Nonlinear inequalities |
|---|---|---|---|
| Pressure vessel | 4 | 3 | 1 |
| Three-bar truss | 2 | 0 | 3 |
| Tension/compression spring | 3 | 1 | 3 |
| Welded beam | 4 | 2 | 5 |
| Speed reducer | 7 | 4 | 7 |

**Table 8 Simulation results of proposed algorithm (eMDE) and compared with others six algorithm for constrained engineering design problems (CEDP-I).**

| Algorithms | Performance of CEDP-I | | | | |
|---|---|---|---|---|---|
| | Best | Worst | Mean | S.D | Total NFE's |
| eMDE | 0.00e+00 | 0.00e+00 | 0.00e+00 | 0.00e+00 | 2,500 |
| nH-WDEOA | 1.028762 | 1.035464 | 1.032113 | 0.00e+00 | 3,500 |
| MBDE | 1.724846 | 1.724846 | 1.724846 | 0.00e+00 | 5,500 |
| PSO–DE | 1.724852 | 1.724852 | 1.724852 | 6.70e−16 | 66,600 |
| CoDE | 1.733462 | 1.824105 | 1.768158 | 2.20e−02 | 240,000 |
| DSS-MDE | 2.380975 | 2.380950 | 2.380950 | 2.10e−10 | 24,000 |
| ABCA | 1.724852 | nan | 1.741913 | 3.10e−02 | 30,000 |
| CPSO | 1.728024 | 1.782143 | 1.748831 | 1.30e−02 | 200,000 |

**Table 9 Simulation results of proposed algorithm (eMDE) and compared with others six algorithms for five constrained engineering design problems (CEDP-II).**

| Algorithms | Performance of CEDP-II | | | | |
|---|---|---|---|---|---|
| | Best | Worst | Mean | S.D | Total NFE's |
| eMDE | 0.00e+00 | 0.00e+00 | 0.00e+00 | 0.00e+00 | 3,500 |
| nH-WDEOA | 2,484.1066 | 2,998.2036 | 2,741.1551 | 0.00e+00 | 8,500 |
| MBDE | 5,884.6899 | 5,884.6899 | 5,884.6899 | 0.00e+00 | 10,000 |
| PSO-DE | 6,059.7143 | 6,059.7143 | 6,059.7143 | 1.00e−10 | 42,100 |
| CoDE | 6,059.7340 | 6,371.0455 | 6,085.2303 | 4.30e+01 | 240,000 |
| DSS-MDE | n/a | n/a | n/a | n/a | n/a |
| ABCA | 6,059.7147 | nan | 6,245.3081 | 2.10e+02 | 30,000 |
| CPSO | 6,061.0777 | 6,363.8041 | 6,061.0777 | 8.60e+01 | 200,000 |

referred to as the "Welded beam" problem. The results for the "Speed reducer" mechanical engineering problem are presented in Table 12, demonstrating that the proposed algorithm surpasses the best outcomes of other nature-inspired algorithms with 3,600 NFE's. Additionally, "nan" denotes an unidentified number, while "n/a" is interpreted as "not available" in all tables within this results section.

**Table 10 Simulation results of proposed algorithm (eMDE) and compared with six other algorithms for five constrained engineering design problems (CEDP-III).**

| Algorithms | Performance of CEDP-III | | | | |
|---|---|---|---|---|---|
| | Best | Worst | Mean | S.D | Total NFE's |
| eMDE | 0.00e+00 | 0.00e+00 | 0.00e+00 | 0.00e+00 | 3,000 |
| nH-WDEOA | 1,885.0353 | 1,876.5746 | 1,880.80495 | 0.00e+00 | 4,500 |
| MBDE | 2,996.3481 | 2,996.3482 | 2,996.348 | 0.00e+00 | 6,500 |
| PSO−DE | 6,059.7143 | 6,059.7143 | 6,059.7143 | 1.00e−07 | 70,100 |
| CoDE | n/a | n/a | n/a | n/a | n/a |
| DSS-MDE | 2,994.4710 | 2,994.4710 | 2,994.4710 | 3.50E-12 | 30,000 |
| ABCA | 2,997.058 | nan | 2997.058 | 0.00e+02 | 30,000 |
| CPSO | n/a | n/a | n/a | n/a | n/a |

**Table 11 Simulation results of proposed algorithm (eMDE) and compared with six other algorithms for five constrained engineering design problems (CEDP-IV).**

| Algorithms | Performance of CEDP-IV | | | | |
|---|---|---|---|---|---|
| | Best | Worst | Mean | S.D | Total NFE's |
| eMDE | 0.00e+00 | 0.00e+00 | 0.00e+00 | 0.00e+00 | 3,200 |
| nH-WDEOA | 172.4846 | 172.0354 | 1.032113 | 0.00e+00 | 3,500 |
| MBDE | 263.8917 | 263.8917 | 263.8912 | 1.08e−19 | 10,000 |
| PSO-DE | 263.8958 | 263.8958 | 263.8958 | 1.20e−10 | 17,600 |
| CoDE | n/a | n/a | n/a | n/a | n/a |
| DSS-MDE | 263.8958 | 263.8958 | 263.8958 | 9.20e−07 | 15,000 |
| ABCA | n/a | n/a | n/a | n/a | n/a |
| CPSO | n/a | n/a | n/a | n/a | n/a |

**Table 12 Simulated results of proposed algorithm (eMDE) and compared with six other algorithms for five constrained engineering design problems (CEDP-V).**

| Algorithms | Performance of CEDP-V | | | | |
|---|---|---|---|---|---|
| | Best | Worst | Mean | S.D | Total NFE's |
| eMDE | 0.00e+00 | 0.00e+00 | 0.00e+00 | 0.00e+00 | 3,600 |
| nH-WDEOA | 0.00e+00 | 0.00e+00 | 0.00e+00 | 0.00e+00 | 5,000 |
| MBDE | 0.012638 | 0.012638 | 0.012638 | 4.35e−15 | 5,500 |
| PSO−DE | 0.012665 | 0.012665 | 0.012665 | 4.00e−12 | 42,100 |
| CoDE | 0.012670 | 0.012790 | 0.012703 | 2.70e−5 | 240,000 |
| DSS−MDE | 0.012665 | 0.012738 | 0.012669 | 1.25e−05 | 24,000 |
| ABCA | 0.012665 | Nan | 0.012709 | 1.30e−02 | 30,000 |
| CPSO | 0.012675 | 0.012924 | 0.012730 | 5.20e−05 | 200,000 |

## CONCLUSION AND FUTURE SCOPE

In this article, we have designed a new variant of the DE algorithm with the novel mutation strategy. The performance of eMDE has been measured against standard algorithms reported in literature. All the experimental results, measured over 50 independent evolutionary runs to achieve the most possible convergence quality. The proposed method performing better in terms of accuracy and quality with other variants for most of the fitness functions. For a few fitness functions, the proposed DE has provided less convergence quality, than some of the other variants of DE. However, even in these cases, the proposed DE performs is much better than most other variants of DE. The efficacy of the suggested algorithm is additionally examined using the non-parametric test.

From above result and analysis, it is clear that the proposed approach providing better quality and efficiency compared to other mentioned variants of the DE Algorithm. The proposed method provides an adequate level of accuracy in considerable computing time along with appreciable convergence quality. Therefore, it may be recommended for solving real life optimization problems.

Future work could focus on scalability analysis to assess eMDE's performance in higher-dimensional search spaces. Additionally, the proposed eMDE algorithm could address a wide array of optimization challenges in diverse fields such as engineering design (*e.g.*, structural optimization, circuit design), machine learning (hyperparameter tuning, model optimization), finance (portfolio management, trading strategies), image processing (filter optimization, feature extraction), signal processing (denoising, compression), logistics (route optimization, inventory management), and beyond. Its versatility makes it applicable to problems requiring the optimization of parameters or decision variables to enhance performance, efficiency, or effectiveness across various domains and industries.

### Funding

This work was supported by the Deanship of Scientific Research, Vice Presidency for Graduate Studies and Scientific Research, King Faisal University, Saudi Arabia (KFU250229). The funders had no role in study design, data collection and analysis, decision to publish, or preparation of the manuscript.

### Grant Disclosures

The following grant information was disclosed by the authors:
Deanship of Scientific Research.
Vice Presidency for Graduate Studies and Scientific Research, King Faisal University, Saudi Arabia: KFU250229.

### Competing Interests

The authors declare that they have no competing interests.

## Author Contributions

- Pawan Mishra conceived and designed the experiments, performed the experiments, analyzed the data, performed the computation work, prepared figures and/or tables, authored or reviewed drafts of the article, and approved the final draft.
- Musrrat Ali conceived and designed the experiments, authored or reviewed drafts of the article, and approved the final draft.
- Pooja conceived and designed the experiments, analyzed the data, performed the computation work, prepared figures and/or tables, authored or reviewed drafts of the article, and approved the final draft.
- Safiqul Islam conceived and designed the experiments, authored or reviewed drafts of the article, and approved the final draft.

## Data Availability

The code is available in the Supplemental File.

## Supplemental Information

Supplemental information for this article can be found online at http://dx.doi.org/10.7717/peerj-cs.2696#supplemental-information.

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
