# Peer review of "Enhanced mutation strategy based differential evolution for global optimization problems"

_PeerJ Computer Science, doi:10.7717/peerj-cs.2696_

## Round 0.1 · original submission · Major Revisions

Please ensure that the comments and concerns raised by the reviewers are addressed in the major revision.

Reviewer 1 ·

Basic reporting

In the study, a new mutation phase has been proposed for the differential evolution algorithm. The proposed method has been tested on 27 different standard benchmark functions. Comparisons with current and widely used differential evolution methods such as JADE, SaDE, CoDE, DSIDE, MPEDE are given during testing. In addition, statistical analyzes were made. But the shortcomings in the article are listed below.

Experimental design

1- Real world problems should be added to demonstrate the applicability of the results of the study (Such as Real World Engineering Design Problem).
2- In Table 2, all the best performances in all benchmark functions should be bold.
3- Comparison with WOA alone is not enough. Why was only WOA chosen? It should be compared with at least 5 different methods such as slime mold algorithm, arithmetic optimization algorithm, archimedes optimization algorithm from current methods.

Validity of the findings

4- In addition to the Convergence curve graph, boxplot graph should also be given.
5- COMPARATIVE STUDY AND DISCUSSION part should be improved. The results should be scientifically explained and discussed.
6- The explanation of benchmark functions is insufficient. 27 different standard benchmark functions have been used. In addition, it is not specified what capabilities the results of the benchmark functions show the algorithms (unimodal, multimodal, composite or hybrid etc.?). These should be explained and the advantages of the proposed method should be stated. Leading publications related to these (Chaotic slime mould optimization algorithm for global optimization, Comparison of current metaheuristic optimization algorithms with CEC2020 test functions) must be cited.
7- Sensitivity analysis should be performed for the proposed method.

Additional comments

8- The word " plateform " in the sentence “The proposed algorithm is implemented with Python programming language in Python 3.0 240 plateform” should be corrected as “platform”. In addition, errors like this should be reviewed throughout the article.
9- The number of iterations should be expressed properly (2000?).
10- The complexity of the proposed method should be given.

Reviewer 2 ·

Basic reporting

The authors present a new variant of the differential evolution method with a new mutation strategy. They tested the performance of the method against recent algorithms and some state-of-the-arts making on 27 benchmark functions.

1- Abstract does not clearly explain the contribution. The motivation of the paper does not exist. The contribution is not properly explained in an understandable way. The abstract section should be rewritten in order to clearly state the manuscript's main focus. The abstract should give the readers essential details, i.e., including the main contributions, the proposed method, the main problem, the obtained results, the benchmark tests, the comparative methods, etc. Efforts are needed to make the abstract coherent while clearly describing the problem being investigated and findings.
2- Some of the variables in the equations listed in the definitions need to be explained. Some mathematical notations are not rigorous enough to correctly understand the contents of the paper. The authors are requested to recheck all the definition of variables and further clarify these equations.
3- Clarifying the study’s limitations allows the readers to better understand under which conditions the results should be interpreted. A clear description of limitations of a study also shows that the researcher has a holistic understanding of his/her study. However, the authors fail to demonstrate this in their paper. The authors should clarify the pros and cons of the methods. What are the limitation(s) methodology(ies) adopted in this work? Please indicate practical advantages, and discuss research limitations.
4- The starting search points of the algorithms are not clear. Are they same for all the algorithms? Have the simulations been performed in the same situations? How do you guarantee a fair comparison?
5- There is no detailed discussion and the authors do not explain why the proposed method is superior.
6- There is not any experiment and analysis about the performance of the proposed method on real engineering problems.
7- Introduction section seems voluminous, broad, and heterogeneous. The authors are supposed to focus on the main topic of the study and present a Literature Review in the form of tables in order to make research gaps and innovations easy to detect. Authoritative synthesis assessing the current state-of-the-art is absent.
8- Table and figure captions should be corrected. The cannot be a sentence such as “Table 5. Statistical analysis of eMDE for results shown in table 2.” Table 2 should be polished.
9- Furthermore, in general, the literature review is not sufficient. It is more of the type “researcher X did Y” rather than an authoritative synthesis assessing the current state-of-the-art. Where do we stand today? What approaches are there in the literature to model the problem? What are the main differences between them? What are their weaknesses and strengths?
10- What do the variables represent? It is not clear how the constraints (for example for decision variables) is handled?
11- The reason for specially selecting the mentioned benchmark functions does not exist.
12- Graphics and charts need more explanation.
13- Some paragraphs are too long to read. The authors should try for readability and comprehensibility by dividing paragraphs into two or more.
14- The current introduction is simple and misses many contents related to the problem formulation. There is not a clear categorization of related works. The intelligent optimization algorithms discussed are simplistic for this topic. The authors listed various optimization methods without any classification and this situation arises confusion. These methods are categorized into nine different classes according to the papers entitled “Plant intelligence based metaheuristic optimization algorithms” and “Comparative Assessment of Light-Based Intelligent Search and Optimization Algorithms”. These papers should be considered by citing in order to prevent confusion.
15- The mutation equation does not seem a “novel” equation and it is used in different adaptive or dynamic metaheuristic methods.
16- There are English grammar, writing style errors, and typos.
17- Additional comments about the reached results should be included.
18- What are the other possible methodologies that can be used to achieve your objective in relation to this work?
19- The conclusion section is indicative, but it might be strengthened to highlight the importance and applicability of the work done with some more in-depth considerations, to summarize the findings, and to give readers a point of reference. The conclusion is confused in general. Concerning Conclusion section, it would be better "Conclusions and Future Research", and it is strongly suggested to include future research of this manuscript. What will be happen next? What we supposed to expect from the future papers? 18. Please expand “Conclusion” section with some specific ideas for future work. Additional comments about the reached results should be included.
20- The values for the parameters of the algorithms selected for comparison are not given.
21- The paper lacks the running environment, including software and hardware. The analysis and configurations of experiments should be presented in detail for reproducibility. It is convenient for other researchers to redo your experiments and this makes your work easy acceptance. A table with parameter setting for experimental results and analysis should be included in order to clearly describe them.

Experimental design

1- The starting search points of the algorithms are not clear. Are they same for all the algorithms? Have the simulations been performed in the same situations? How do you guarantee a fair comparison?
2- There is no detailed discussion and the authors do not explain why the proposed method is superior.
3- There is not any experiment and analysis about the performance of the proposed method on real engineering problems.
4- What do the variables represent? It is not clear how the constraints (for example for decision variables) is handled?
5- The reason for specially selecting the mentioned benchmark functions does not exist.
6- Additional comments about the reached results should be included.
7- The paper lacks the running environment, including software and hardware. The analysis and configurations of experiments should be presented in detail for reproducibility. It is convenient for other researchers to redo your experiments and this makes your work easy acceptance. A table with parameter setting for experimental results and analysis should be included in order to clearly describe them.

Validity of the findings

1- The mutation equation does not seem a “novel” equation and it is used in different adaptive or dynamic metaheuristic methods.
2- What are the other possible methodologies that can be used to achieve your objective in relation to this work?
3- The values for the parameters of the algorithms selected for comparison are not given.

---

## Round 0.2 · Major Revisions

Please submit a revision where the comments of the reviewer are suitably addressed, especially regarding the comments on experimental design and validity of the findings. For value comparisons with recent work, please cite more relevant literature within the last five years. Also please justify why the comparison against WOA is a valid comparison instead of more recent methods.

Reviewer 2 ·

Basic reporting

1- Equations should be used with correct equation number. Please do not use “given as”, “as follows”, etc. Explanation of the equations should also be checked. All variables should be written in italic as in the equations. Their definitions and boundaries should be defined. Necessary references should be given.
2- Many of the equations are part of the related sentences. Attention is needed for correct sentence formation.
3- Literature review seems weak. The works presented in Literaure Review section seem old. More recent papers especially on differential evolution variants and dynamic (adaptive) stratiges embedded to the optimizationm methods should be analyzed.

Experimental design

1- Experimental results are weak. There are many differential evolution variants proposed recently. The compared algorithms are too old for optimization research. Recent literature should be deeply explored and more comprehensive results should be provided.
2- Results from real engineering problems are not presented. Furthermore more complex and benchmark functions should be used.

Validity of the findings

1- Comparison with WOA is not enough to make a general conclusion. The works should also be compared different, recent, and "real novel" optimization algorithms.

---

## Round 0.3 · Minor Revisions

Please address the minor revisions requested by the Reviewer2. The manuscript will be accepted if these minor revisions are addressed.

Reviewer 2 ·

Basic reporting

Many of the previous comments and questions are properly addressed. However

- Some sentences are too long to read. They may be divided into two or more.
- Pay special attention to the usage of abbreviations. Spell out the full term at its first mention, indicate its abbreviation in parenthesis and use the abbreviation from then on.
- Equations are still not used with correct equation number in the revised paper. Many of the equations are part of the related sentences. Attention is needed for correct sentence formation.

Experimental design

All queries and issues are duly addressed, and the requisite additions and modifications are implemented.

Validity of the findings

All queries and issues are duly addressed, and the requisite additions and modifications are implemented.

Additional comments

Should these minor amendments be implemented, I would be grateful if you would refrain from issuing a further invitation for me to review and revise the document.

---

## Round 0.4 · accepted · Accept

I confirm that I have assessed the revision myself, and I am happy with the current version